# Psychological Correlates of Ghosting and Breadcrumbing Experiences: A Preliminary Study among Adults

**DOI:** 10.3390/ijerph17031116

**Published:** 2020-02-10

**Authors:** Raúl Navarro, Elisa Larrañaga, Santiago Yubero, Beatriz Víllora

**Affiliations:** Department of Psychology, Faculty of Education and Humanities, University of Castilla-La Mancha, Avda de los Alfares, 42, 16071 Cuenca, Spain; elisa.larranaga@uclm.es (E.L.); santiago.yubero@uclm.es (S.Y.); beatriz.villora@uclm.es (B.V.)

**Keywords:** ghosting, breadcrumbing, dating relationships, adults, satisfaction with life, loneliness, helplessness

## Abstract

The present study aimed to examine differences in three psychological constructs (satisfaction with life, loneliness, and helplessness) among adults experiencing ghosting and breadcrumbing. A sample of 626 adults (303 males and 323 females), aged from 18 to 40 years, completed an online survey asking to indicate whether someone they considered a dating partner had ghosted or breadcrumbed them in the last year and to complete three different scales regarding satisfaction with life, loneliness, and helplessness. The results showed than those participants who had indicated experiencing breadcrumbing or the combined forms (both breadcrumbing and ghosting) reported less satisfaction with life, and more helplessness and self-perceived loneliness. The results from the regression models showed that suffering breadcrumbing would significantly increase the likelihood of experiencing less satisfaction with life, and of having more feelings of loneliness and helplessness. However, no significant relation was found between ghosting and any of the examined psychological correlates.

## 1. Introduction

Online dating has drastically changed the dating scenario since it was launched 20–25 years ago. Homosexual and heterosexual men and women have included online dating platforms into their lives to search for romantic and sexual relationships. The mean age of these users fluctuates between 24 and 31 years old, although dating apps are becoming popular with other populations like seniors (>65 years). However, with young people (13–18 years old), the Internet has not yet substituted in-person encounters [1]. Dating apps offers access to more likely dates and sexual encounters, allows online communication with potential partners, helps to acquire information about people before meeting them in person, and offers diverse tools to negotiate stages of their love/sex relationships [2,3]. Nevertheless, dating apps also have disadvantages like the gamification of relationships, engaging in risky behaviors (e.g., unprotected sex, disclosing personal information, stalking and cyberstalking, sexual victimization) and being exposed to behaviors like “breadcrumbing”, “slow fading”, “benching”, “haunting”, or “ghosting” [4,5]. These behaviors are examples of how people use Internet-mediated communication and dating apps to flirt, initiate, maintain, or end relationships. However, very few published studies have examined these phenomena, and research about their potential correlates among those who has suffered these online behaviors is scarce. The primary aim of the present study was to examine the psychological correlates of two digital behaviors (breadcrumbing and ghosting) in the emerging and young adults who have suffered them.

### 1.1. Ghosting and Breadcrumbing: Definition and Research

“Ghosting” originates from the noun “ghost”. According to the Cambridge Dictionary, ghosting means “a way of ending a relationship with someone suddenly by stopping all communication with them” [6]. Ghosting refers to “unilaterally access to individual(s) prompting relationship dissolution (suddenly or gradually) commonly enacted via one or multiple technological medium(s)” [7]. Ghosting occurs through one technological means or many by, for example, not responding to phone calls or text messages, no longer following partners or blocking partners on social network platforms. Ghosting differs from other relationship dissolution strategies insofar as it takes place without the ghosted mate immediately knowing what has happened, who is left to manage and understand what the partner’s lack of communication means [8] and is unable to close the relationship [9]. Ghosting prevalence has been examined mostly in US adults. Prevalence rates range between 13% and 23% for those adults who have been ghosted by a romantic partner [8,10]. In Spain, 19.3% have reported having suffered ghosting at least once in the past year [11].

Empirical evidence for ghosting behaviors is extremely scarce. Ghosting has been conceptualized by past research, which describes it as a strategy adopted to dissolve undesired relationships without ever having to break them up [9]. Other studies have investigated which factors are associated with ghosting. The relation of ghosting with implicit theories was analyzed by Freedman et al. [8], who found that the participants reported a more frequent acceptability of ghosting, more ghosting intentions, and ghosting being used more in the past. These authors also reported firmer destiny beliefs (i.e., steady and invariable relationships). Koessler, Kohut, and Campbell [12] discovered that the relationships which ended via ghosting were more short term and characterized by less commitment than those terminated by direct conversation. Navarro et al. [11] revealed that ghosting behaviors are linked to using online dating sites/apps, the time spent on online dating apps/sites, online surveillance, and more short-term relationships.

“Breadcrumbing” originates from the noun “breadcrumbs”, which means “very small pieces of dried bread, especially used in cooking” [13]. Breadcrumbing is defined as “the act of sending out flirtatious, but non-committal text messages (i.e., “breadcrumbs”) to lure a sexual partner without expending much effort” or “when the “crush” has no intentions of taking things further, but they like the attention. So they flirt here or there, send DMs/texts just to keep the person interested, knowing damn well they’re staying single” [14]. Breadcrumbers do not stop calling definitely, but sporadically send text messages or DMs, give the occasional wink or post likes on social networks like Instagram just frequently enough for receivers to not lose interest in them, but not enough for relationships to develop. Breadcrumbing is not such a clear dissolution strategy as ghosting because, despite breadcrumbing possibly occurring when a relationship terminates, the initiator does not wish to let his/her partner go. It is also a way of keeping a date on “hold” and a form of social dynamics in which breadcrumbers are not really attracted to the other person, but are interested in remaining relevant/attractive for others [15]. Empirical evidence for breadcrumbing is more limited than that for ghosting. Navarro et al. [11] reported a 35.6% prevalence rate of Spanish adults who stated that they had been victims of breadcrumbing. These authors also found that breadcrumbing was linked with employing online dating sites/apps, more short-term relationships and online surveillance of people met online. However as far as we know, no research has studied the psychological correlates related to experiencing breadcrumbing or ghosting.

### 1.2. Psychological Correlates of Ghosting and Breadcrumbing

The quality of our close relationships is the biggest predictor of longevity and happiness [16], and quality intimate relationships are associated with fewer mental health problems and better subjective well-being [17]. On the contrary, poor quality intimate relationships and breakups are often associated with less well-being, such as anger, sadness, psychological distress, and depression [18]. Research has shown that social media and technology have extended our social network and our number of interpersonal interactions [19]. However, moving our relationships to digital environments and the growing use of digital technologies are also associated with less depth in our connections, increased loneliness and less satisfaction with life [4,20,21]. Consequently, a rise in acts such as breadcrumbing and ghosting could increase the rates of loneliness, hopelessness and lack of satisfaction with life in those who suffer them.

Research has not yet examined the consequences of experiencing ghosting and breadcrumbing victimization. Popular media have echoed different opinions by psychologists and sociologists who warn about the harmful consequences of these types of online strategies to maintain or dissolve relationships. In an interview for The Huffington Post, Sherry Turkle, a professor from MIT, expressed that “ghosting has serious consequences, because when they treat us as if we could be ignored, we begin to think that this is fine and we treat ourselves as people who don’t have feelings. And at the same time, we treat others as people who have no feelings in this context, so empathy begins to disappear” [22]. On the Psychology Today website, Jennice Vilhauer claimed that ghosting can have a serious impact on a person’s mental health. She explained that “ghosting is the ultimate use of the silent treatment, a tactic that has often been viewed by mental health professionals as a form of emotional cruelty. It essentially renders you powerless and leaves you with no opportunity to ask questions or be provided with information that would help you emotionally process the experience. It silences you and prevents you from expressing your emotions and being heard, which is important for maintaining your self-esteem” [23].

In her article “Breadcrumbing Is the New Ghosting and It’s Savage”, the journalist Samantha Swantek explained that “breadcrumbing can be especially infuriating if you’re in search of a genuine connection. Breadcrumbers waste your time and introduce a sense of falsehood into a rapport that you may have conceived as real” [24]. During an interview for The New York Times, Professor Turkle explained that online connections where breadcrumbing takes place “can have the paradoxical effect of making the person who receives them feel let down rather than gratified” [25]. Although research has not explicitly examined the psychological correlates of ghosting and breadcrumbing victimization, it can be hypothesized that people experiencing ghosting and breadcrumbing victimization will also experience negative psychological correlates if we consider that these digital tactics can reinforce their insecurities and affect their future relationships.

No research has examined the possible consequences of ghosting and breadcrumbing, but we can pay attention to research that examines the consequences of a distinct, but related, phenomenon: ostracism. Ostracism may include being ignored by an individual or a group in a wide range of contexts, such as friendship networks, family, workplace and on the Internet [26]. The negative consequences of ostracism have been widely documented. Ostracism threatens fundamental needs, such as belonging, self-esteem, control, and meaningful existence and can, consequently, increase loneliness, depressed mood, frustration, anxiety and helplessness [27]. Research has found that being ignored over the Internet (cyberostracism) is related to psychological distress, emotional dysregulation, loneliness, sadness and anxiety [26,28,29,30].

Ghosting is a way to end relationships, whereas breadcrumbing can act as a way to maintain certain relationships for different purposes. However, as the people experiencing them may be awaiting answers (text messages, phone calls, likes, or responses to posted messages on social networks), lack of responses (with ghosting), or sporadic responses (with breadcrumbing) may place someone in a standby mode that might trigger the feeling of being ostracized [31]. When individuals feel ostracized on the Internet as a result of a relationship breaking up or relationships not progressing, individuals can react with psychological discomfort, including a range of feelings like social pain, loneliness, and distress.

### 1.3. The Present Study

The above literature review demonstrates the importance of studying the psychological correlates of ghosting and breadcrumbing victimization. However, apart from studies on ostracism and relationship breakups [32,33,34], no other study has examined the link between psychological discomfort and digital tactics to end or maintain intimate relationships. This article is the first research to examine the relation among self-reported ghosting victimization, breadcrumbing victimization, and psychological constructs in emerging and young adults. The present study aimed to examine differences in three psychological constructs (satisfaction with life, loneliness, and helplessness), previously found to be correlated with different forms of ostracism, among adults exposed to ghosting and breadcrumbing victimization, and puts forward the following research hypothesis:

**Hypothesis** **1 (H1).***Differences in satisfaction with life, loneliness, and helplessness scores are expected among groups of participants experiencing ghosting and breadcrumbing (ghosting victims, breadcrumbing victims and combined victims) and non-involved participants. We also expect to find more loneliness and helplessness and less satisfaction with life in the combined victims group. This hypothesis was based on past research showing that the detrimental effects of negative events may differ depending on the nature of the behavior suffered [35]. Research has also reported that people exposed to several ostracism events are more likely to report poorer mental health outcomes, such as depression and anxiety [27]*.

**Hypothesis** **2 (H2).***Ghosting and breadcrumbing are expected to be associated with less satisfaction with life and more feelings of loneliness and helplessness after controlling for demographics variables. Previous research has found that negative events, such as online aggressive behaviors or partner phubbing, are associated with lower levels of subjective well-being and psychological distress [36,37]. Based on this previous evidence, we expect to find the same relation with other online behaviors like ghosting and breadcrumbing victimization*.

## 2. Methods

### 2.1. Study Design and Participants

An online cross-sectional survey with young adults was conducted in Spain in 2019. Convenience and snowball sampling was the method followed to recruit those who participated herein. First, the URL to the anonymous online survey was forwarded to PhD students from a medium-sized university in central Spain. Students were requested to forward the link to the survey to their family members and any acquaintances who they knew had Internet access, a mobile phone, and one short-term or long-term relationship or more irrespectively of their present sentimental status. The survey was answered by 716 young people aged between 18 and 40. Of the initial sample, 64 people were ruled out for reporting never having been engaged in a short-term or long-term relationship. An instructional manipulation check was used to verify that the participants had read the survey instructions. Following Oppenheimer, Meyvis, and Davidenko [38], it consisted of a question embedded within the survey scales that was similar to the other questions in length and response format. However, unlike the other questions, the IMC asked participants to ignore the standard response format and instead provide a confirmation that they had read the instruction. Twenty-six participants failed this check and were removed before running analyses. The final sample was made up of 626 participants (mean age = 29.64 years; SD = 8.84). An equivalent number of females (*n* = 323) and males (*n* = 303) filled in the online survey, of whom 79.4% had finished Higher Education. Moreover, 82.9% stated being heterosexual, and 17.1% were lesbian, gay, or bisexual. According to their dating history, the participants had been engaged in two relationships on average (M = 2.23, SD = 0.74), with a range of one to six relationships. Finally, 390 participants (62.3%) stated that they had a relationship with their partner when they answered the online survey.

### 2.2. Procedures

Those participants who clicked the link to the online survey were requested to complete a self-administered questionnaire after giving informed consent. All the participants were able to assess the questionnaire once. The respondents’ anonymity was guaranteed. The Clinical Research Ethics Committee of the Virgen de la Luz Hospital in Cuenca approved the study protocol (PI0519). All the subjects signed informed consent forms prior to participating in the study.

### 2.3. Measurements

**Demographics.** The participants reported their age, gender, sexual orientation, level of education, current relationship status, and number of relationships according to their dating history.

**Ghosting experiences.** As “ghosting” is Anglo-Saxon, the online survey specified this definition to allow the participants to be familiar with it: “ghosting” refers to unilaterally ceasing all communication (temporarily or permanently) with someone with whom some kind of relationship is maintained (friendship or sentimental). It is a way to break up a relationship (sudden or gradual) in which all contact with that person is cut off, or their attempts to communicate with the person who started it are ignored. Commonly, “ghosting” occurs through one or technological means or many; for example, not responding to phone calls or WhatsApp messages, ceasing to follow or blocking social networks sites. After the definition, the participants were asked to indicate whether someone they considered a dating partner had ghosted them in the last year. Items scored on a 5-point scale: 0 (*never*); 1 (*not in the last year, but before*); 1 (*once or twice*); 3 (*3 to 5 times*); 4 (*more than 5 times*).

**Breadcrumbing experiences.** As “breadcrumbing” is Anglo-Saxon, the online survey specified this definition to allow the participants to be familiar with it: “breadcrumbing” literally refers to leaving bread crumbs so that someone can follow the trail. Breadcrumbers do not stop talking on WhatsApp, sending random DMs or text messages, or giving an occasional like on a social network site in order to not discard the other person at all, but the relationship does not progress. Breadcrumbing can happen after a relationship has broken up, but the initiator does not wish to let the partner go. It is also a way to maintain a date on “hold” and a type of social dynamics in which breadcrumbers are not really attracted to the other person, but are interested in remaining relevant/attractive to others. After the definition, the participants had to state if someone they considered a dating partner had breadcrumbed them in the past year. Items scored on a 5-point scale as follows: 0 (*never*); 1 (*not in the last year, but before*); 2 (*once or twice*); 3 (*3 to 5 times*); 4 (*more than 5 times*).

**Satisfaction with life.** The employed scale was the Satisfaction with Life Scale developed by Diener et al. [39]. For this study, the Spanish validation by Cabañero-Martínez et al. [40] was utilized, namely a 5-item self-report measure designed to assess global cognitive judgments of one’s satisfaction with life. The response format is a 5-point Likert scale ranging from 1 = strongly disagree to 6 = strongly agree. An example of an item on this scale is “So far I have gotten the important things I want in life”. Reliability in the current sample yielded a McDonald’s omega of 0.88.

**Helplessness.** It was examined with the Multidimensional Fatalism Scale developed by Esparza et al. [41]. This subscale consists in six descriptive self-statements with a rating response ranging from 1 (*not at all*) to 5 (*very much*). An example of an item on this scale is “There’s nothing I can do to succeed in life, since one’s level of success is determined when one is born”. Scores were obtained by summing items, with better scores indicating a higher degree of helplessness beliefs. In the present study, McDonald’s omega reliability coefficient was 0.87.

**Loneliness.** It was examined by the short 8-item version of the UCLA Loneliness Scale [42]. An example of an item on this scale is “I often feel alone”. Items scored on a Likert scale ranging from 0 (*Never*) to 3 (*Often*)*,* where higher scores indicated a stronger endorsement of loneliness. McDonald’s omega reliability coefficient was 0.91 herein.

### 2.4. Data Analysis

The percentages of adults’ self-reported ghosting and breadcrumbing experiences were first computed. Second, differences in ghosting and breadcrumbing according to gender, sexual orientation, and relationship status were analyzed by chi-square tests. Third, an analysis of variance (ANOVA) was performed to examine the differences in the three psychological constructs (satisfaction with life, helplessness, loneliness) among the three established groups of victimization experiences (ghosting victims, breadcrumbing victims, combined victims). Finally, multiple regression analyses were employed to determine the association among ghosting, breadcrumbing and the three psychological constructs by adjusting for gender, age, sexual orientation, and relationship status. In order to better analyze possible age differences, following previous research [1,43], age was dichotomized categorizing participants into emerging adults (18–25 years old) and young adults (26–40 year old). The SPSS 24 statistical software was used for all the analyses with the exception of McDonald’s omega coefficient that was estimated through the free software JASP (Version 0.11.1) [44].

## 3. Results

### 3.1. Preliminary Analysis: Frequency Distribution of Ghosting and Breadcrumbing Experiences

Participants’ classification as victims of ghosting, breadcrumbing, or the combination of both these digital tactics, was made by following a highly restrictive criterion used by previous research about other forms of digital teasing [45]. Participants were considered ghosting victims if they reported experiencing it at least 3 times or more in the last year. Participants were considered breadcrumbing victims if they reported having suffered it 3 times or more in the past year. Those participants who reported having suffered both ghosting and breadcrumbing 3 times or more were considered victims of both digital tactics (combined victims). The remaining participants were considered to be not involved (see Table 1). This procedure resulted in 30 (4.8%) individuals being categorized as ghosting victims, 88 (14.1%) as breadcrumbing victims, 15 (2.4%) as ghosting and breadcrumbing victims, and 493 (78.8%) as not involved.

No significant differences in ghosting according to gender, sexual orientation, age or current sentimental status (single or have a partner) were found among the participants. However, single participants reported experiencing significantly more breadcrumbing than the participants with a partner (χ^2^ = 22.173, *p <* 0.001). No significant differences in breadcrumbing were found according to gender, sexual orientation and age.

### 3.2. ANOVA Analysis: Examining Differences in Psychological Constructs (Satisfaction with Life, Helplessness, and Loneliness) among the Digital Tactics Used to End or Maintain Relationships (Ghosting, Breadcrumbing, and the Combined Form)

An ANOVA was carried out to examine the differences in the psychological constructs among victims of ghosting, of breadcrumbing, and the victims of the combination of these two digital tactics (see Table 2). The pair-wise differences in the adjusted means were examined by a *post hoc* test with Bonferroni correction.

A significant difference was found in the degree of satisfaction with life, helplessness, and loneliness among the four groups (victims of ghosting, breadcrumbing, combined abuse, and those not involved). With Bonferroni *post hoc* test analyses, the not involved and ghosting victims groups appeared to be more satisfied with life and presented the lowest level of helplessness and loneliness compared to the breadcrumbing victims and the combined victims.

### 3.3. Multiple Regression Analyses: Testing the Relation between Psychological Constructs and Self-Reported Ghosting and Breadcrumbing

To examine the associations between the different strategies to end or maintain relationships and satisfaction with life, helplessness and loneliness, a multiple regression analysis was conducted after adjusting for gender, age, sexual orientation, and current relationship status using the enter method. Collinearity diagnostics that are available on SPSS 24.0 were conducted. An examination of Tolerance and Condition Index figures revealed that multicollinearity was not a problem since the lowest levels of the former were 0.71 and figures for the latter did not exceed 21.0. The results are presented in Table 3. Gender and sexual orientation were not associated with satisfaction with life, helplessness and loneliness. Only age was significantly associated with satisfaction with life. Those participants aged 26 years or older reported higher satisfaction with life rates than the participants aged less than 26 years. Current relationship status was significantly associated with the three psychological constructs. Single participants reported less satisfaction with life, and more loneliness and helplessness than those participants with a partner. No significant relation was found between suffering ghosting and any of the examined well-being dimensions. On the contrary, a positive relation appeared between the breadcrumbing and psychological constructs. The participants who obtained higher breadcrumbing rates also reported lower levels of satisfaction with life, but higher levels of helplessness and loneliness. Nevertheless, the relation between breadcrumbing and helplessness was weak.

## 4. Discussion

The objective of the present work was to analyze the psychological correlates of two digital strategies used to end or maintain sentimental relationships among adults, specifically known as breadcrumbing and ghosting. We initially expected to find: (1) differences in satisfaction with life, helplessness, and loneliness between those who had suffered these strategies and those who had not, and worse scores for those who had suffered both types of strategies; (2) suffering ghosting and breadcrumbing would increase the likelihood of informing less satisfaction with life, and more helplessness and loneliness, after controlling for the demographic variables.

However, our hypotheses were only partly confirmed. Those participants who had indicated experiencing breadcrumbing or the combined forms (both breadcrumbing and ghosting) reported less satisfaction with life, and more helplessness and self-perceived loneliness. In the same line, the results from the regression models showed that suffering breadcrumbing would significantly increase the likelihood of experiencing less satisfaction with life, and of having more feelings of loneliness and helplessness. Surprisingly, however, no significant relation was found between ghosting and any of the examined psychological correlates. In fact, the participants who reported having suffered ghosting in the past 12 months obtained similar means for satisfaction with life, loneliness, and helplessness than those who had experienced neither ghosting nor breadcrumbing (not involved). These results contradict those found by former research into the associations on mental health of not only sentimental breakups, but also of different forms of ostracism [29,34].

Nonetheless, the results of the present study did not necessarily indicate that the people suffering ghosting do not experience psychological discomfort, but are able to recover from the ended relationship and, thus, improve their satisfaction with life, as well as their feelings of loneliness and helplessness. According to the Temporal Need-Threat Model of ostracism (TNTM [27]), getting over an event that involved someone being ignored depends on the extent to which this event has threatened our social requirements, the time the event lasted, the number of events suffered and individual characteristics.

For the analyses performed in the present study, we selected those participants who had suffered ghosting and breadcrumbing 3 times or more in the last year. The purpose was to analyze those cases with fewer possibilities of recovering because ghosting and breadcrumbing had been recurrent in the 12 months prior to them answering the survey about these practices. However, in this scenario, a significant association with the three studied psychological correlates was found for only breadcrumbing. A possible explanation of the association between breadcrumbing and the psychological correlates is that breadcrumbing may operates in a similar way that addictive behaviors. According to the Incentive Theory [46,47], the basic motivator of behavior in the anticipation of the reward. In this sense, breadcrumbing could have reinforcing properties. It is possible that the expectation generated by the breadcrumbing experienced (random messages, likes, comments in social networks), the anticipation of a possible reward for these behaviors together with their lack of predictability (it is not possible to predict when the reinforcement will be received), generate tension on those who suffer breadcrumbing and, in consequence, arouse negative psychological repercussions. In view of our findings, future research should analyze if the unpredictability that characterized breadcrumbing is more harmful than relationship dissolution conducted through ghosting.

The lack of associations between ghosting and the psychological correlates examined can have several explanations. One possible explanation for this lack of association could be that, although the participants who informed about more ghosting events were selected, these events could have taken place further back in time. So any immediate or short-term effects would have disappeared. Repeated ghosting events did not necessarily lead to having developed learnt helplessness, but could have contributed to generate better coping strategies for ghosting. Conversely, as breadcrumbing is a phenomenon that can continue with time, and one that can lead to those suffering it being in a “continuous situation of being on hold”, it might generate more emotional tension, less satisfaction with life and could, therefore, make recovery from an ongoing process difficult as its effects continue with time. Future research must analyze to what extent previous breakups have been overcome or experiencing similar situations could contribute to learn coping strategies that minimize the negative effects of these online strategies.

Individual characteristics can also explain the lack of associations between ghosting and psychological correlates. Research into cyberostracism reveals that the immediate effects of ostracism (e.g., feelings of loneliness) can come into play regardless of individual characteristics [27]. After a time however, individual characteristics like fewer rumiative tendencies [48], less social anxiety [49], or traits like psychological flexibility [26], are associated with a greater ease with which to recover from psychological distress suffered as a result of ostracism. These results highlight the need to analyze what other individual variables can mitigate the negative effects of ghosting or which ones intervene when facilitating the recovery of those suffering this practice. For example, different research works have found that resilience is a relevant variable for recovering from different online conducts [50,51]. Resilience, and other variables identified by research about ostracism, must be analyzed by future research to know if they cushion effects of ghosting, and to see if they also moderate the effects found for breadcrumbing.

Moreover, research about breakup adjustment difficulties reveals that the effects of breakups depend on how expected the breakup was, the duration and the type of relationship, and the breakup intensity [18]. Although the present study does not report such information, it is likely that lack of significant associations between ghosting and psychological correlates is related to the type of sentimental relationship. Former research has revealed that ghosting is a more widely used dissolution strategy in short-term relationships characterized by less commitment [12]. Long-term relationships are associated with suffering more distress after breaking up than short-term ones [52]. Future research must analyze the impact of ghosting on mental health according to the type of relationship that ended via ghosting (e.g., long-term, short-term, hook-ups). It is likely that those who have invested more time and emotional resources in a relationship may suffer more emotional distress, especially if they did not expect the breakup [37,53]. Given its characteristics, although ghosting is an unexpected and surprising event for whoever suffers it, it might not have the same effects on relationships involving more commitment or those which lasted longer than on those involving less commitment. Relationships affected by breadcrumbing do not progress and are never well-established, thus they lack commitment (at least for those who started breadcrumbing). However, those who experience breadcrumbing remain in a “standby” state with time, which can often make victims feel excluded. So, compared to ghosting, it can be suffered as a more intense ostracism experience, which is why it has more negative effects on mental health.

The last explanation for lack of relations between ghosting and psychological correlates is that, as with different types of aggression practiced over the Internet, practices like ghosting are becoming common, and may be rationalized and minimized by those experiencing them. Indeed, research has demonstrated that perceived harm or subjective experience associated with various forms of aggression is related to lack of negative effects on the mental health of those suffering it [54,55]. Qualitative research is desirable to examine adults’ perceptions of and beliefs in ghosting, and why they were unable to perceive it as being as harmful as breadcrumbing.

### Limitations in the Present Study

This study has several limitations that ought to be taken into account when interpreting its results. First, participants were recruited through both convenience sampling (students) and snowball sampling. Consequently, the generalizability of the results remains unclear given that participants could choose to pass the survey link to individual that share the same characteristics. Future research should use more effective approaches to reach and recruit the sample such as random sampling to avoid biased generalization of data. Second, as all the data were collected by cross-sectional self-report measures that were obtained online, we were unable to make causal interpretations and we cannot be sure that the participants provided accurate reports about their online behaviors. Third, we measured both ghosting and breadcrumbing by asking only one question and we did not ask about the time that had elapsed since these practices started or the kind of relationship maintained with someone who practiced ghosting or breadcrumbing. Future research should collect more detailed data about such behaviors and the type of relationship that was ended by ghosting or maintained by breadcrumbing (e.g., casual sexual meetings, short-term relationships, committed relationships). Fourth, this study only analyzed the relation of ghosting and breadcrumbing victimization with three psychological correlates (satisfaction with life, helplessness, and loneliness) previously linked to other forms of ostracism. Future research must analyze other mental health-related variables like depression, anxiety, etc., as well as antisocial behaviors such as aggression. Finally, although the age range is relatively wide, acquiring information from other age groups would be interesting. Future research should include people younger than 18 years and older than 40 years because they may also be exposed to such behaviors.

## 5. Conclusions

The present study contributes to increase our knowledge about the digital tactics used to end and maintain relationships by means of Internet-mediated communication, namely two limited investigated phenomena: ghosting and breadcrumbing. It also helps to improve our understanding of the possible correlates of the psychological health of those experiencing them, and offers new directions to study what emotional impact these practices have as part of sentimental relationships among adults. As no significant relations were found between ghosting and the analyzed psychological constructs, this study draws researchers’ attention to analyze which other mental health-related variables may be related with these behaviors.

## Figures and Tables

**Table 1 ijerph-17-01116-t001:** Summary statistics of the study variables and frequency distribution of ghosting and breadcrumbing experiences (N = 626).

Study Variables	
Psychological correlates	*M (SD)*
Satisfaction with life	3.40 (0.88)
Helplessness	2.12 (0.89)
Loneliness	5.1 (0.81)
Digital tactics to end or maintain relationships	% (*n*)
Not involved	78.8 (493)
Ghosting	4.8 (30)
Breadcrumbing	14.1 (88)
Combined victims (Ghosting and Breadcrumbing)	2.4 (15)

Note: Values represent the means and standard deviations of the continuous variables, and the percentages and number of the participants of the categorical variables.

**Table 2 ijerph-17-01116-t002:** Differences in the psychological constructs among victims of ghosting, breadcrumbing, combined victims, and those not involved.

Variables	Digital Tactics	N	M (SD)	F	η^2^	Bonferroni Test
Satisfaction with life	Ghosting	30	3.43 (0.51)	4.961 ***	0.04	Not, Gh > Brd, Com
	Breadcrumbing	88	3.19 (1.03)			
	Combined	15	2.78 (0.96)			
	Not involved	493	3.46 (0.85)			
Helplessness	Ghosting	30	2.23 (0.75)	6.134 ***	0.08	Not, Gh > Brd, Com
	Breadcrumbing	88	2.35 (1.00)			
	Combined	15	2.82 (1.09)			
	Not involved	493	2.05 (0.85)			
Loneliness	Ghosting	30	5.14 (0.83)	8.336 ***	0.06	Not, Gh > Brd, Com
	Breadcrumbing	88	5.30 (0.89)			
	Combined	15	5.96 (0.78)			
	Not involved	493	5.04 (0.77)			

*** *p <* 0.001. (Not = not involved; Gh = Ghosting; Brd = Breadcrumbing; Com = Combined ghosting and breadcrumbing).

**Table 3 ijerph-17-01116-t003:** Regression analyses examining the associations of ghosting and breadcrumbing with psychological constructs as the criterion.

Variables	Satisfaction with Life			Helplessness			Loneliness		
	B	SE	β	B	SE	β	B	SE	β
Gender ^a^	0.105	0.066	−0.060	−0.119	0.070	−0.068	0.000	0.060	−0.000
Age ^b^	−0.011	0.004	−0.108 **	0.001	0.004	0.014	−0.002	0.004	−0.020
Sexual Orientation ^c^	−0.062	0.092	−0.026	0.076	0.097	0.032	0.065	0.084	0.030
Relationships status ^d^	0.619	0.073	0.340 ***	−0.227	0.077	−0.123 ***	−0.561	0.067	−0.335 ***
Ghosting	−0.079	0.130	−0.023	0.244	0.137	0.071	0.190	0.118	0.061
Breadcrumbing	−0.194	0.093	−0.081 **	0.324	0.055	0.143 **	0.170	0.084	−0.078 **
R2 (Adj. R2)	0.129 (0.120)			0.046 (0.037)			0.148 (0.140)		
F	150.216 ***			50.017 ***			170.936 ***		

Note. N = 643. B = Beta coefficient not standardized; S.E = Standard Error; β = Beta coefficient standardized. ^a^ 0 = female, 1 = male; ^b^ age (1 = 18-25 years, 2 = 26-40 years); ^c^ 0= heterosexual, 1 = lgb; ^d^ 0 = single, 1 = with a partner; ** *p <* 0.01; *** *p <* 0.001.

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
