# Peer review of "Psychological Correlates of Ghosting and Breadcrumbing Experiences: A Preliminary Study among Adults"

_ijerph, 2020, doi:10.3390/ijerph17031116_

Round 1

Reviewer 1 Report

I think investigating breadcrumbing, ghosting, etc is important to do but I especially found interest in the connections between life satisfaction and these online behaviors, however breadcrumbing is not always directly tied to sexual liason expectations, but a more broad range of levels of intimacy. With that said, results tell a story that makes sense in regard to the behavior. 

Author Response

We appreciate you took time to perform the review of our manuscript.

I think investigating breadcrumbing, ghosting, etc is important to do but I especially found interest in the connections between life satisfaction and these online behaviors, however breadcrumbing is not always directly tied to sexual liason expectations, but a more broad range of levels of intimacy. With that said, results tell a story that makes sense in regard to the behavior. 

Authors’ answer: thank you for your comment and your kind consideration. We agree with you. For this reason we have include alternative explanations considering the Incentive Theory described in page 9.

Reviewer 2 Report

METHOD
The sampling technique used (snowball) has numerous limitations that should be considered and expressed by the authors as one of the limitations of the research.

The authors state that: "An instructional manipulation check was used to verify that the participants had read the survey instructions" ... Please clarify what exactly is this control method?

The description of the final sample indicates the average age of the men and the average age of the women? And in the total sample?
In subsequent analyzes, the dichotomized age variable is used, using a 25-year cut-off point. Explain the procedure and criteria used to use this value and not another, has any statistical analysis been applied for this? What are they based on? This should be clearly explained in the text.

ANALYSIS OF DATA
They should indicate the procedure for estimating the reliability of the scales. In this case, the authors use McDonald's Omega coefficient. Therefore, they should specify whether they have used a macro for SPSS or another type of resource as a specific spreadsheet for their estimation. In addition, they must include the quotation in the text and its corresponding reference.

RESULTS
In table 3, they must indicate the meaning of B, SEB, ... in the lower note of the table.

There is no case (table 3) of * p <0.05, therefore, this annotation should be deleted.

Indicate the method used to compute the regression model (enter, stepwise, ...). No data on collinearity tests between the variables are provided.

DISCUSSION
There are references that are first introduced in the discussion, which are not in the introduction. They should know that the Discussion section tries to contrast the results obtained with respect to the hypotheses raised and in relation to the studies that have previously been mentioned in the introduction.

Author Response

First of all, thank you for your revision and you valuable comments for the improvement of our manuscript.

METHOD
The sampling technique used (snowball) has numerous limitations that should be considered and expressed by the authors as one of the limitations of the research.

Authors’ answer: thank you for your comment. We agree and we have included the sampling method as a limitation of the study. Page 10.

The authors state that: "An instructional manipulation check was used to verify that the participants had read the survey instructions" ... Please clarify what exactly is this control method?

Authors’ answer: we have now described in what consisted the IMC. The information has been included in page 4.

The description of the final sample indicates the average age of the men and the average age of the women? And in the total sample?

Authors’ answer: this was a typo. Instead of written “mean age” we wrote “males”. We have corrected the mistake. Sorry about that.

In subsequent analyzes, the dichotomized age variable is used, using a 25-year cut-off point. Explain the procedure and criteria used to use this value and not another, has any statistical analysis been applied for this? What are they based on? This should be clearly explained in the text.

Authors’ answer: You are right. In order to better understand possible differences according to age. We dichotomized age in two categories according to what previous research and systematic reviews has called emerging adults and young adults. We split the sample according to the ages included in the two categories. The dichotomization is now explained in page 5-6 (data analyses section).

ANALYSIS OF DATA
They should indicate the procedure for estimating the reliability of the scales. In this case, the authors use McDonald's Omega coefficient. Therefore, they should specify whether they have used a macro for SPSS or another type of resource as a specific spreadsheet for their estimation. In addition, they must include the quotation in the text and its corresponding reference.

Authors’ answer: you are right. We have include now the citation to the JASP software used.

RESULTS
In table 3, they must indicate the meaning of B, SEB, ... in the lower note of the table.

Authors’ answer: thank you. We have include a note wit the meaning of the statistics.

There is no case (table 3) of * p <0.05, therefore, this annotation should be deleted.

Authors’ answer: thank you. The annotation has been deleted.

Indicate the method used to compute the regression model (enter, stepwise, ...). No data on collinearity tests between the variables are provided.

Authors’ answer: compute method is now indicated and information abouth the collinearity test is now reported. Please, see page 7

DISCUSSION
There are references that are first introduced in the discussion, which are not in the introduction. They should know that the Discussion section tries to contrast the results obtained with respect to the hypotheses raised and in relation to the studies that have previously been mentioned in the introduction.

Authors’ answer: we agree with you. However, given that some of the results were unexpected, together with references included in the introduction section we had to include new references in order to try to offer an explanation of the unexpected findings.

Reviewer 3 Report

there is the need to clarify in all the introductory part that

1.we are talking about effects on victims. we are not talking about the breadcrumber and the ghoster. There may be some confusion in the reader-

2.in the introducion and in the abstract there is the need to clarify that the article will  explain how breadcrumbing  and ghosting will be detect. This concept  is clearly defined in the research description, but not in the introduction 

3. line 37 "potential effect", is too strong. The article is not concern about consequences, but  about potential connected feeling of loneliness ... 

4. AROUND LINE 53 THE READER NEED TO KNOW THAT THE RESEARCHER WILL , IN THE NEXT PARAGRAPH, PROPOSE A CLEAR DEFINITION OF BOTH THESE BEHAVIOURS  (breadcrumbing and ghosting) IN THE EXPERIENCE OF THE INTERVIEWED

5.line 134 CLARIFY that the article is concern about BEING VICTIMS OF BREADCRUMBING

6. line 383  sattisfaction with life is not a direct measure of potential ostracism

the article present a description of two style of closing relations being more and more frequent thanks the online connection tools. 

however the 2 words ghosting and breadcrumbing are still belonging to a very specialized audience. I was aware about the phenomenon but not about these words. Therefore their presentation  should  be clearly done at the beginning of the article;  Dictionary reference should be introduced from the beginning of the article. The reader should not wait the final part of the text. 

Author Response

Thank you for you kind and valuable review. Bellow are our answers to your comments.

1.we are talking about effects on victims. we are not talking about the breadcrumber and the ghoster. There may be some confusion in the reader-

Authors’ answer: thank you. We have now clarify this possible confusion in the entire document adding the word victimization, victims or explaining that we examined people suffering it.

2.in the introducion and in the abstract there is the need to clarify that the article will  explain how breadcrumbing  and ghosting will be detect. This concept is clearly defined in the research description, but not in the introduction 

Authors´ answer: thank you. We have described in the abstract how ghosting and breadcrumbing were measured.

line 37 "potential effect", is too strong. The article is not concern about consequences, but  about potential connected feeling of loneliness ... 

Authors´ answer: you are right. We have modified that sentence.  

AROUND LINE 53 THE READER NEED TO KNOW THAT THE RESEARCHER WILL , IN THE NEXT PARAGRAPH, PROPOSE A CLEAR DEFINITION OF BOTH THESE BEHAVIOURS  (breadcrumbing and ghosting) IN THE EXPERIENCE OF THE INTERVIEWED

Authors´ answer: thank you. We have now included a clear definition of both phenomena in the introduction.

5.line 134 CLARIFY that the article is concern about BEING VICTIMS OF BREADCRUMBING

Authors’ answer: thank you. We have now clarify this possible confusion in the entire document adding the word victimization, victims or explaining that we examined people suffering it.

line 383  sattisfaction with life is not a direct measure of potential ostracism

Author’s answer: You are right. There was a typo in that sentence and we have now corrected it. Sorry for the confusion.

however the 2 words ghosting and breadcrumbing are still belonging to a very specialized audience. I was aware about the phenomenon but not about these words. Therefore their presentation  should  be clearly done at the beginning of the article;  Dictionary reference should be introduced from the beginning of the article. The reader should not wait the final part of the text. 

Authors´ answer: thank you. Following your advice. We have now included a clear definition of both phenomena in the introduction.

Round 2

Reviewer 2 Report

Thank you for attending to my recommendations on the manuscript.

Author Response

Thank you for you review and your kind consideration. The manuscript is now better described after your recommendations.